# Climate resilience through bioeconomy: A mixed-methods protocol for assessing adaptation policies in rural settlements on the Amazon

Daniel Silva[1]*, Larissa Alves[2‡], Naurinete Reis[3‡], Maclem Erane Gonçalves dos Santos[3‡], Emilio Mendes[4‡], Vitor Castro[5‡]

1 Institute for Agrarian and Regional Development Studies, Federal University of Southern and Southeast Pará, Pará, Brazil, 2 Institute of Economics, State University of Campinas, São Paulo, Brazil, 3 Center for Affirmative Actions, Federal University of Southern and Southeast Pará, Pará, Brazil, 4 Institute of Applied Social Sciences, Federal University of Pará, Pará, Brazil, 5 Institute of Geosciences, Federal University of Southern and Southeast Pará, Pará, Brazil

☙ This author contributed equally to this work.
‡ LA, NR, ME, EM and VC also contributed equally to this work.
* daniel.nogueira@unifesspa.edu.br

## Abstract

### Introduction

Climate change intensifies social, economic, and environmental vulnerabilities in Amazonian rural communities, where dependence on natural resources and limited institutional capacity constrain adaptation. The bioeconomy is often proposed as an alternative pathway for sustainable development, yet empirical evidence of its role in resilience remains scarce.

### Objective

This study protocol presents a mixed methods design to evaluate climate adaptation policies and community resilience strategies in agrarian reform settlements, rural territories created through Brazil's land redistribution program, in Pará, Brazil. Although centered on a regional case, the protocol is structured to generate insights applicable to other socioecologically vulnerable rural contexts in the Amazon and comparable regions worldwide.

### Methods

The study will be conducted in 15 municipalities across the three regional superintendencies of the National Institute for Colonization and Agrarian Reform. The design integrates (i) secondary data on socioeconomic, climatic, and policy indicators; (ii) semi-structured interviews with municipal managers (n = 1–3 per municipality); (iii) focus groups with community leaders (n = 8–12 per settlement); and (iv) household

**Data availability statement:** No datasets were generated or analysed during the current study. All relevant data from this study will be made available upon study completion.

**Funding:** This research was funded by the National Council for Scientific and Technological Development (CNPq), Brazil, through Call 19/2024 – Pro Amazônia, grant number 445058/2024-2 awarded to DS. The funder had no role in the design of the study; in the collection, analyses, or interpretation of data; in the writing of the manuscript; or in the decision to publish the results.

**Competing interests:** The authors declare no competing interests. The study received public research funding from the National Council for Scientific and Technological Development (CNPq), Brazil (Grant No. 445058/2024-2), which had no role in study design, data collection and analysis, decision to publish, or preparation of the manuscript.

surveys (n = 20–50 families per settlement). Primary outcomes include the Income and Livelihood Diversification Index (ILDI), Climate Vulnerability Perception Scale (CVPS), Bioeconomy Engagement Index (BEI), and Reported Adaptive Capacity Index (RACI). Secondary measures include the Adaptation Strategy Diversity Index (ASDI) and a Municipal Climate Engagement Typology (MCET). Quantitative analyses will employ multilevel linear and generalized linear models (LMM/GLMM), while qualitative data will be analyzed through reflexive thematic analysis, with final integration achieved via a triangulation framework.

## Expected results

This study will generate a comprehensive framework integrating indicators of vulnerability, adaptive capacity, and bioeconomy engagement across agrarian reform settlements in the Eastern Amazon. By combining household surveys, qualitative fieldwork, and municipal-level policy analysis, the protocol provides a transparent, replicable approach for assessing local climate adaptation processes. The study is designed to inform municipal and state policymakers, support the strengthening of local adaptation initiatives, and contribute evidence relevant to Sustainable Development Goals (SDGs 2, 13, and 15).

## Introduction

Climate change represents one of the main threats to sustainable development, especially for vulnerable rural communities in regions of high ecological sensitivity such as the Amazon [1]. These populations face complex challenges in which poverty, limited access to public policy programs, and dependence on fragile production systems interact with environmental impacts, exacerbating their vulnerability [2,3]. The intensification of extreme events, including prolonged droughts and atypical floods, has already compromised the livelihoods of communities dependent on family farming and natural resources [4].

In this context, the bioeconomy has emerged as a potential pathway to reconcile economic development and environmental conservation, relying on the sustainable use of biodiversity and the valorization of traditional knowledge [5]. However, its implementation faces critical challenges, including conceptual ambiguity – where differing definitions may or may not generate socio-environmental benefits [6] – and the pace of climate change, which may limit the feasibility of economic model's dependent on ecosystems already under pressure [7].

Recent scholarship emphasizes that advancing the Amazonian bioeconomy requires not only technological innovations but also multi-scalar policy changes and institutional support [6]. From a sustainability transitions perspective, the effective scaling of socio-bioeconomy initiatives depends on improvements in infrastructure, value chains, and social organizations, which remain unevenly developed across region [8]. Research on non-timber forest products – such as açaí, Brazil nut, and

babassu – demonstrates their essential role in linking biodiversity conservation with livelihood generation. Yet, while production, distribution, and economic aspects are relatively well documented, legal, regulatory, and socio-political frameworks remain underexplored [9]. Moreover, the interaction of rural families with forests extends beyond economic considerations, encompassing cultural, subsistence, and spiritual dimensions, which complicates the straightforward promotion of bioeconomy value chains. Together, these insights highlight the need for a comprehensive, context-sensitive approach that integrates economic, ecological, and social perspectives.

Parallel to these debates, studies show that Amazonian communities are already devising local adaptation strategies to climate variability and extreme events. Research using a Community – Based Adaptation lens has documented a wide range of coping, adaptive, and transformative responses to extreme floods, despite limited governmental support [10]. Other work shows how floodplain residents integrate natural hydrological and ecological processes into their management systems, optimizing ecosystem functioning through traditional knowledge [11]. Similarly, farmers in various regions have adopted new crop varieties, irrigation systems, and strategic land-use practices to enhance resilience. Yet they continue to face barriers such as water scarcity, high input costs, and insufficient technical assistance, underscoring the importance of institutional support and access to sustainable technologies [12].

These findings demonstrate the capacity of rural communities to devise locally grounded adaptation responses, but they also reveal persistent structural challenges, particularly weak integration of local strategies into formal public policies. Importantly, while evidence on adaptation is expanding, a critical gap remains in understanding how these processes unfold in agrarian reform settlements, which represent one of the largest rural populations in the Amazon and sit at the intersection of climate vulnerability, land reform, and bioeconomy development.

Despite growing efforts to assess climate impacts and adaptive responses in Amazonian rural settings, methodological approaches remain fragmented and unevenly applied. This protocol contributes to these ongoing discussions by integrating climate adaptation, resilience, and bioeconomy dimensions into a coherent multilevel framework [13]. This gap constrains the design of effective public policies, which often rely on generic indicators that do not capture the social, ecological, and institutional specificities of vulnerable rural populations. Likewise, although the bioeconomy is frequently promoted as a development pathway, evidence on its effectiveness under rapidly changing climatic conditions remains limited and uneven [14].

This protocol seeks to address these questions by combining the assessment of climate adaptation and mitigation policies with the analysis of bioeconomy chains in agrarian reform settlements. In Brazil, *agrarian reform settlements* (*assentamentos de reforma agrária*) are rural territories created and regulated by the National Institute for Colonization and Agrarian Reform (INCRA), where land is redistributed to landless or land-poor families as part of the federal land reform policy. Families receiving plots through this program are commonly referred to as *settled families* or *beneficiary families*.

The results will contribute not only to filling local data gaps [15] but also to advancing global debates on climate adaptation [16], bioeconomy [17], and sustainable development [18], directly engaging with the Sustainable Development Goals (SDG 2 – Zero Hunger, SDG 13 – Climate Action, and SDG 15 – Life on Land).

## Materials and methods

### Objectives, hypotheses, and research strategies

This study protocol was developed with the aim of identifying and analyzing public policies and community strategies for climate change mitigation and adaptation in municipalities and rural communities in the Eastern Brazilian Amazon. The specific research objectives are: (i) To map and quantify climate vulnerability through the construction and analysis of secondary indicators of exposure, sensitivity, and adaptive capacity. (ii) To identify, through field research, the perceived effects of climate change, adaptation and mitigation strategies, and the main barriers faced by small-scale family farmers in selected rural settlements. (iii) To map municipal and state-level public policies for addressing climate change, including an analysis of the public budget allocated to these actions. (iv) To analyze the perception, implementation challenges, and articulation of climate policies from the perspective of local public managers.

This is an observational, cross-sectional study. Therefore, the analyses are designed to identify robust statistical associations and to explore potential causal mechanisms, but they cannot definitively establish causality due to the potential for unmeasured confounding and reverse causality. To enhance transparency and reduce researcher degrees of freedom, we pre-specify a concise set of primary outcomes and hypotheses (detailed in Table 1 and the 'Primary and Secondary Outcomes' section). All other analyses will be clearly labeled as secondary or exploratory.

The approach integrates public policy indicators (e.g., existence of mitigation and adaptation plans, climate budget) with community perceptions, allowing for a broader analysis of ongoing impacts and adaptation strategies. The data collected under this protocol aims to test hypotheses based on (i) spatial patterns; (ii) socioeconomic patterns; (iii) demographic patterns; (iv) institutional/political patterns; and (v) bioeconomy/climate patterns.

Table 1 summarizes the main hypotheses guiding this work.

**Primary and secondary outcomes.** To reduce researcher degrees of freedom and enhance transparency, this protocol pre-specifies four primary outcomes directly linked to the hypotheses presented in Table 1: (i) the Income and Livelihood Diversification Index (ILDI, linked to PH2), (ii) the Climate Vulnerability Perception Scale (CVPS, linked to PH1), (iii) the Bioeconomy Engagement Index (BEI, linked to PH5), and (iv) the Reported Adaptive Capacity Index (RACI, linked to PH4). All other indicators are classified as secondary and will be used for exploratory analysis. The conceptual definitions are presented below, while the detailed computational procedures and statistical validation strategies are provided in the section *Operationalization of the Indices*.

Based on the hypotheses presented in Table 1, this protocol pre-specifies a set of primary and secondary outcomes to ensure analytical parsimony, clarity, and replicability.

Primary outcomes (core indicators of household resilience and bioeconomy engagement):

1. Income and Livelihood Diversification Index (ILDI): a continuous score (0–1) that synthesizes the distribution of agricultural, extractive, and non-agricultural income. This index is adapted from standard diversification measures widely used in livelihood and rural development research (e.g., Herfindahl–Hirschman approaches [30]). *Expected direction*: higher values will be associated with greater diversification of adaptation actions (PH2).

**Table 1. Hypotheses guiding the protocol.**

| Analyzed Pattern | Primary Hypothesis (PH) | Theoretical Premise |
|---|---|---|
| **Spatial and Environmental** | PH1: Settlements with greater forest cover and ecological connectivity are expected to be associated with higher adaptive capacity, more diverse of bioeconomy practices, and a lower perception of climate vulnerability. | Intact ecosystems provide more ecosystem services (provisioning, regulating) that function as buffers against climate impacts and as a foundation for resilient economic strategies [19–21] |
| **Socioeconomic and Livelihoods** | PH2: Household income diversification (agricultural, extractive, non-agricultural) is positively associated with higher perceived climate resilience and greater adoption of adaptation actions. | Diversification reduces dependence on a single activity vulnerable to climate events, enabling the absorption of shocks and the reallocation of resources [22,23] |
| **Demographic and Social Capital** | PH3: Multigenerational demographic composition and participation in community organizations are positively correlated with the adoption of a more diversified portfolio of adaptation strategies (both traditional and innovative). | Intergenerationality facilitates the transmission of traditional knowledge, while social organization facilitates access to information, resources, and external policies [24,25]. |
| **Institutional and Public Policies** | PH4: The existence and implementation of participatory public policies on climate and bioeconomy explicitly at the municipal level are associated with higher adaptive capacity reported by communities and greater productive diversification based on socio-biodiversity. | Well-designed and participatory policies provide the resources, incentives, and structures necessary for communities to implement effective adaptation strategies at scale [26,27]. |
| **Bioeconomy and Climate** | PH5: Engagement in bioeconomy value chains (particularly processing and certification) is positively associated with higher household income levels and the adoption of climate mitigation and adaptation practices. | Value addition and access to differentiated markets create economic incentives for conservation and for investment in sustainable and resilient practices [28,29]. |

2. Climate Vulnerability Perception Scale (CVPS): a reflective scale (1–5) constructed from Likert-type items assessing perceived frequency, severity, and impacts of climate-related events. This scale is adapted conceptually from prior work on climate risk perception but operationalized specifically for Amazonian rural contexts. Internal consistency and dimensionality will be assessed through Cronbach's alpha [31] and confirmatory factor analysis. *Expected direction*: lower perceived vulnerability in settlements with greater forest cover and ecological connectivity (PH1).

3. Bioeconomy Engagement Index (BEI): an ordinal variable (level 0 = no engagement; 1 = primary production; 2 = processing; 3 = processing + certification). This is an original indicator developed by the authors, grounded in literature on value-chain upgrading and socio-bioeconomy transitions. *Expected direction*: Higher values reflect greater involvement in bioeconomy value chains. This indicator will be used to examine whether higher engagement is associated with household income and with adoption of adaptation and mitigation practices (PH5).

4. Reported Adaptive Capacity Index (RACI): a reflective score derived from farmers' self-assessment of their ability to respond to climate impacts. The index is constructed as the average of Likert-scale items (range 1–5), where higher values indicate greater perceived adaptive capacity. Items draw on established adaptation assessment frameworks but were adapted to the Amazonian rural context. This indicator is used to examine associations proposed in PH4.

Households with multigenerational demographic composition and participation in community organizations (PH3) will report higher adaptive capacity (as measured by the RACI), and this association will be supported by a more diversified portfolio of adaptation strategies (ASDI). This hypothesis is grounded in demo-livelihoods frameworks [32], which emphasize the role of intergenerational structures and social capital in shaping livelihood decisions and adaptive responses.

Secondary outcomes (exploratory measures to enrich analysis):

a) Adaptation Strategy Diversity Index (ASDI): a count-based index capturing the number and variety of adaptation practices reported by each household, including both traditional (e.g., collective work, soil management, water storage) and innovative strategies (e.g., improved seeds, irrigation systems, shade management). This is an author-developed indicator, conceptually grounded in livelihood and adaptation portfolio approaches widely used in climate adaptation research. The index reflects the breadth of a household's adaptation repertoire, with higher values indicating a more diversified set of responses to climate impacts. The ASDI supports the examination of PH3 by providing a complementary measure of how demographic composition and social capital relate to adaptation strategy diversity.

b) Municipal Climate Engagement Typology: a contextual variable (high, medium, or low engagement) constructed from a systematic analysis of municipal climate-related policies and budgets. The typology follows consolidated methodologies [33,34], combining two dimensions: (i) *policy engagement*, assessed through the presence, depth, and quality of municipal adaptation and mitigation plans (including objectives, instruments, and implementation mechanisms); and (ii) *budgetary engagement*, assessed through the classification and quantification of climate-relevant expenditures in municipal budgets. Municipalities are then categorized into high, medium, or low engagement based on their combined policy and budget scores (used to test HP4).

All outcomes will be operationalized on continuous, ordinal, or categorical scales, as appropriate, and incorporated into multilevel linear and generalized linear models (LMM/GLMM), with families (level 1) nested within settlements (level 2), which are nested within municipalities (level 3).

A complete description of how each indicator is computed, including variable selection, scaling, aggregation, and validation procedures, is provided in the section Operationalization of the Indices.

**Data collection methods.** The study employs three complementary data collection methods: household surveys, semi-structured interviews, and focus groups. The household survey captures quantitative information on socioeconomic characteristics, productive activities, climate perceptions, and adaptation practices, using a structured questionnaire administered face-to-face to a randomly selected adult decision-maker in each household.

Qualitative data will be collected through two instruments. First, semi-structured interviews will be conducted with municipal public managers (1–3 per municipality), allowing in-depth exploration of policy implementation challenges, resource constraints, and perceptions of local climate risks. Second, focus groups with community leaders (8–12 participants per settlement) will be used to capture collective experiences, shared vulnerabilities, and community-level adaptation strategies. Both methods are designed to generate rich contextual information that complements survey findings.

Detailed guides for each instrument (survey, interview, and focus group) are provided in the Supplementary Material (S1–S3 Files).

**Ethical considerations.** This protocol was approved by the Research Ethics Committee of the Federal University of Pará (CAAE: 87492325.6.0000.0018). All participants will provide informed consent prior to data collection. Identifiable information will be removed during transcription and data processing, and all results will be reported in aggregated form to ensure confidentiality.

## Sample, inclusion and exclusion criteria, and participant characteristics

The study will be conducted in the state of Pará, in the Eastern Brazilian Amazon. The selection of this state as the study area is strategically grounded in its centrality to debates on agrarian development, environmental conservation, and climate justice in the Brazilian Amazon [35].

Critically, Pará concentrates approximately 25% of all agrarian reform settlements in the country, hosting about 27% of all settled families in Brazil, representing the largest absolute population in agrarian reform projects within a single state. This demographic density of settlements, located in the Amazon biome – which is globally recognized as one of the most climate-sensitive – creates a unique and crucial socioecological context for investigation [36].

Focusing on this state allows the examination, on a microscale, of the challenges and adaptation strategies of a vast and vulnerable population, whose experiences are directly shaped by the intersection of land tenure policies, deforestation pressures, and the intensification of climate extremes. Therefore, the results generated from this sample will not only have significant local relevance but will also provide transferable insights to other regions of the Pan-Amazon facing similar dynamics, directly contributing to filling evidence gaps in climate adaptation and mitigation policies grounded in social equity.

To operationalize research in this unique context, the study will be conducted in fifteen municipalities of Pará, strategically distributed across the three regional superintendencies of the National Institute for Colonization and Agrarian Reform (INCRA).

The choice of 15 municipalities is not arbitrary but rather designed to capture the diverse socioecological contexts of agrarian reform settlements in Pará. These municipalities encompass distinct territorial and economic features that are central to understanding adaptation dynamics in the Amazon. They include: (i) areas strongly influenced by mining activities, where extractive industries create both opportunities and vulnerabilities; (ii) regions dominated by agribusiness expansion, with intensive land-use change and pressure on family farming; (iii) floodplain and dryland environments, which present contrasting climate risks and livelihood strategies; (iv) municipalities intersected by major highways such as the Transamazon Highway, where infrastructure development shapes access to markets and exposure to environmental impacts; and (v) extractive and conservation-oriented settlements, where socio-biodiversity chains and traditional livelihoods play a central role.

By sampling across this range of contexts, the study captures key socioecological environments characteristic of Amazonian rural settlements. The selection is not numerically balanced across all contexts; rather, municipalities were purposively chosen to ensure representation of major settlement types and territorial dynamics relevant to climate adaptation. This approach enhances analytical robustness and supports the transferability of findings.

In each selected municipality, up to two agrarian reform settlements will be sampled, ensuring representation of the main settlement modalities nationally recognized. This approach will enable a robust comparative analysis of how different

institutional designs and production bases mediate community vulnerability and adaptive capacity in the face of climate change.

**Selection of settlements.** Eligible settlements must: (i) have between 20 and 500 settled families, ensuring comparability and logistical feasibility; (ii) reflect production and socioeconomic conditions representative of the municipality; and (iii) be in a stage of creation, installation, structuring, or consolidation. Settlements with very particular or atypical conditions, such as experimental areas or arrangements that are not replicable, will be excluded.

**Participant characteristics.** Participants include: (i) local public managers (1–3 per municipality), responsible for municipal policies on agriculture, environment, or rural development; (ii) community leaders (8–12 per settlement), invited to participate in focus groups, representing different social and productive segments; and (iii) settled families (up to 50 per settlement), covering a diversity of socioeconomic, productive, and demographic profiles. The study expects to include family farmers across different age groups, education levels, family compositions, and modes of engagement in local markets, as well as varying degrees of dependence on natural resources and participation in public policies. In each household, the respondent will be the adult member primarily responsible for agricultural and livelihood decisions (typically the main decision-maker), as this person is expected to have the most comprehensive knowledge of production, income sources, and climate-related impacts.

This sampling strategy seeks to balance territorial representativeness, social diversity, and logistical feasibility, enabling comparisons across municipalities, settlement types, and family profiles.

**Selection of families.** In each selected settlement, up to 50 families will participate in the study, chosen through simple random sampling from official lists provided by local associations or INCRA. The inclusion criteria for families are: (i) residing in the settlement; and (ii) having an adult household member available to respond to the questionnaire; and (iii) providing voluntary informed consent (ICF). Families will be excluded if: (i) they do not meet the residency requirement; (ii) they refuse or cannot provide consent; or (iii) communication barriers make the interview unfeasible.

**Qualitative saturation criterion.** Qualitative data collection will follow the principle of theoretical saturation [37], assessed iteratively by the analysis team during coding. As a guiding strategy, we will consider saturation reached when two or more consecutive interviews or focus groups no longer generate new relevant codes. However, given the high heterogeneity of Amazonian settlements (e.g., floodplains, mining zones, forest-based communities), this criterion will be applied flexibly. Data collection may be extended in specific contexts if new themes continue to emerge or if settlement-specific dynamics require further exploration. To ensure robustness, saturation will be evaluated both within settlement types and across municipalities and will be triangulated with survey findings and field observations to confirm completeness.

**Sample justification and statistical power.** The study was designed to include 15 municipalities, up to 30 settlements, and 600–1,000 families. Power calculations were conducted to determine the minimum sample size needed to achieve at least 80% power at a 5% significance level for detecting modest-to-moderate associations in a multilevel framework. Because key drivers of climate resilience and bioeconomy engagement (e.g., market access, institutional presence, ecological context) operate predominantly at the community level, statistical power is primarily determined by the number of settlements rather than the number of families per settlement.

To reflect this hierarchical structure, simulations incorporated a range of plausible intraclass correlation coefficients (ICCs = 0.05, 0.10, 0.15, 0.30), informed by the literature [38] and pilot data. All simulations assumed a fixed sample of 600 families distributed across 30 settlements (average cluster size ≈ 20). Models were specified to match our primary outcomes: continuous outcomes (e.g., ILDI) were evaluated using linear mixed models (LMM) with random intercepts at the settlement level, while ordinal outcomes (e.g., the four-level BEI) were analyzed using cumulative link mixed models (CLMM) under a proportional-odds assumption.

Simulation results indicate that the Minimum Detectable Effect (MDE) size is highly sensitive to the degree of within-settlement correlation (Table 2). For low levels of clustering (ICC = 0.05), the study is adequately powered to detect

**Table 2. Simulation-based minimum detectable effects (MDEs) by intraclass correlation coefficient (ICC), assuming N = 600 families, K = 30 settlements, α = 0.05, and 80% power.**

| ICC | MDE (continuous outcome) | MDE (ordinal outcome, log-odds) | MDE (ordinal outcomes, odds ratio) |
|---|---|---|---|
| 0.05 | 0.34 | 0.35 | 1.42 |
| 0.10 | NA | NA | NA |
| 0.15 | 0.47 | 0.50 | 1.65 |
| 0.30 | NA | NA | NA |

moderate effects for both continuous outcomes (MDE ≈ 0.34, in standardized units) and ordinal outcomes (minimum detectable odds ratio ≈ 1.42). Under moderate clustering (ICC = 0.15), detectable effects increase substantially, with an MDE of approximately 0.47 for continuous outcomes and a minimum detectable odds ratio of approximately 1.65 for ordinal outcomes. For higher levels of clustering (ICC ≥ 0.10), no effect sizes within the considered range reached 80% power, indicating that only very large effects would be detectable under such conditions with the planned number of clusters.

These results demonstrate that the proposed sample is well suited to detecting effects of practical relevance under low to moderate intraclass correlation, while also providing a transparent benchmark for interpreting potential null findings in more highly clustered settings. Consistent with this sensitivity analysis, if logistical constraints arise during fieldwork, priority will be given to maximizing the number of settlements rather than increasing the number of families per settlement, as this yields greater power for between-cluster comparisons. Full simulation scripts will be made available in the project's OSF repository to ensure computational transparency and reproducibility [38].

## Description of all processes, interventions, and comparisons

Data collection will be carried out sequentially in two integrated phases, as detailed below. The methodological design was refined following a pilot study conducted in the municipality of Marabá (PA), with the resulting adjustments described to ensure process transparency.

**Phase 1: Secondary data and political-institutional context analysis.** This phase will focus on building a comprehensive diagnosis of the municipal context. It will involve, first, Climate Risk Assessment, through the compilation and analysis of secondary climate indicators widely recognized in the literature (e.g., temperature, precipitation, extreme events) from sources such as the National Institute of Meteorology (INMET) and the National Institute for Space Research (INPE), to characterize the exposure and sensitivity of the sampled municipalities. Second, Public Policy Mapping, through the identification and documentary analysis of municipal and state-level public policies related to climate change, agriculture, and rural development. This will include the analysis of master plans, laws, and decrees, with attention to both actions explicitly labeled as "climate-related" and sectoral actions with indirect impacts on adaptation and mitigation (e.g., sanitation, civil defense, family farming) [26]. Third, Public Budget Audit, applying a data mining methodology to municipal transparency portals [33,34], which allows categorization and quantification of public expenditures directly or indirectly related to the climate agenda, thus overcoming the limitation of expenditures formally tagged as "climatic" [39].

**Phase 2: Primary data collection with key stakeholders.** This phase will capture the perceptions, experiences, and strategies of local actors through three instruments: (i) Household Surveys: administration of a structured questionnaire to a sample of families in each settlement, collecting socioeconomic, productive, climate perception, and adaptation strategy data (S1 File); (ii) Focus Groups with Community Leaders in each settlement type: group discussions to obtain a consolidated view of vulnerabilities, conflicts, and collective adaptation strategies in the settlements, as well as to validate contextual information (S2 File); (iii) Semi-Structured Interviews with Public Managers: interviews with municipal technicians and secretaries to understand perceptions, implementation challenges, and the articulation of climate policies from the perspective of local government (S3 File).

**Post-pilot methodological adjustments.** Testing the instruments in Marabá (PA) resulted in crucial refinements. For Phase 1, the policy and budget analysis methodology was expanded to systematically capture implicit climate actions, as the pilot revealed that most municipal climate responses are driven by sectoral policies (e.g., agriculture, infrastructure) not explicitly labeled as climate related. For Phase 2, the household survey was adapted to include specific modules for riverine and floodplain communities, capturing climate impacts (e.g., changes in flood cycles, river-based access). The interview guide with managers was revised to replace technical jargon with more accessible terms, reflecting the fact that many municipalities lack a consolidated formal framework for climate policy.

**Comparative analysis strategy.** The analysis will be structured across four comparative levels, allowing the isolation of factors associated with resilience at different scales: (i) Across municipalities, contrasting indicators of vulnerability, climate budget, and policy robustness; (ii) Across settlement types, examining how different land tenure and institutional arrangements mediate convergences of families within settlements, correlating socioeconomic profiles (education, income diversification) with adoption of adaptation practices; (iv) Across social actors, triangulating the perspectives of public managers and communities to identify convergences and divergences in risk perceptions and policy effectiveness.

This multilevel approach is essential for identifying robust spatial, socioeconomic, and institutional patterns associated with climate resilience.

## Data analysis plan

Data analysis will follow a sequential explanatory mixed-methods strategy (QUAN→QUAL) [40], in which the initial quantitative analysis will provide an overall picture to be subsequently deepened and contextualized by qualitative investigation. Triangulation of data from different sources (quantitative, qualitative, and secondary) will be employed to validate and enrich the findings, ensuring a robust understanding of the factors that influence climate resilience [41]. All analyses will be performed using R software (version 4.3.0 or higher) for the quantitative component and NVivo software (version 14 or higher) to support the organization and coding of qualitative data.

**General Strategy and Integration.** The analysis will occur in three main stages, corresponding to the data sets, culminating in an integration phase: (i) Secondary Data and Municipal Context Analysis, to characterize the macro context. (ii) Quantitative Primary Data Analysis (Survey), aimed at identifying patterns and testing hypotheses at household and community levels. (iii) Qualitative Primary Data Analysis (Interviews and Focus Groups), to explore causal mechanisms, perceptions, and nuances. (iv) Finally, integration and triangulation, with a synthesis of the findings to answer the research objectives.

**Secondary data and municipal context analysis.** The analysis of secondary data will proceed in three integrated steps to construct key municipal-level variables. First, a Municipal Climate Vulnerability Index (MCVI) will be created by compiling secondary indicators of exposure, sensitivity, and adaptive capacity from sources including AdaptaBrasil, MCTI, IBGE, and INPE. These indicators will be harmonized by converting to per capita measures where appropriate, standardized (z-scores), and aggregated through Principal Component Analysis (PCA) [42]. Second, municipal climate change plans will be assessed using categorical content analysis to generate a plan quality score. Third, a public expenditure audit, employing text mining of municipal transparency portals, will be conducted to identify and quantify climate-related budgets.

To ensure robustness, a multi-step strategy will address data quality challenges. For the MCVI, discrepancies or missing values will be handled using imputation techniques, with sensitivity analyses conducted to assess the impact on the index. Furthermore, all secondary data – including the budget analysis and plan documentation – will be cross-checked against municipal transparency portals and triangulated with qualitative data from interviews and focus groups to ensure contextual accuracy.

Finally, the plan quality scores, and budget allocations will be combined to create a Municipal Climate Engagement Typology (e.g., High, Medium, Low). Both the MCVI and this engagement typology will be incorporated as contextual variables in the multilevel models.

**Quantitative primary data analysis (survey).** Initially, descriptive statistics (frequencies, means, medians, and standard deviations) will be used to characterize the sample profile. Subsequently, to operate key concepts, composite indices will be constructed, and their internal consistency will be assessed using Cronbach's alpha (α > 0.7) and/or Confirmatory Factor Analysis (CFA) [43]. The planned indices include Income and Livelihood Diversification Index (ILDI); Climate Adaptation Practices Index (CAPI); Bioeconomy Engagement Index (BEI); and Climate Vulnerability Perception Scale (CVPS).

To test the hypotheses in Table 1, linear mixed-effects models (LMM) and generalized linear mixed-effects models (GLMM) [44] will be applied, which are appropriate for hierarchical data (families nested within settlements and municipalities). The MCVI and the municipal engagement typology will be included as higher-level variables (level 2), while family characteristics (level 1) will be used as predictors and control variables.

**Addressing confounding and sensitivity analysis.** To mitigate potential confounding, the multilevel models will include a rich set of pre-specified covariates at different levels: (1) Household Level (Level 1): Age and education of the household head, household size, gender of the household head, asset index, length of residence in the settlement; (2) Settlement Level (Level 2): Settlement type, distance to the nearest paved road, presence of a producers' cooperative, years since settlement establishment; (3) Municipality Level (Level 3): Municipal Climate Vulnerability Index (MCVI), Municipal Climate Engagement Typology, and geographic region (e.g., floodplain vs. upland).

Acknowledging the impossibility of measuring all potential confounders, we will conduct a formal sensitivity analysis for the key associations related to our primary hypotheses (e.g., the association between BEI and household income). We will use the sensemakr package in R, following the framework developed by Cinelli and Hazlett (2025) [45]. For each key finding, we will report the Robustness Value (RV), the minimum strength of association (partial $R^2$) that an unmeasured confounder would need to have with both the exposure and the outcome to explain away the observed effect. We will also present contour plots to visualize how the effect estimate and its confidence interval would change under different scenarios of confounding. This will provide readers with a transparent assessment of the robustness of our inferences to unobserved confounding.

**Operationalization of the indices.** To ensure transparency and replicability, each index will be constructed according to clear theoretical and statistical criteria, as follows:

- Income and Livelihood Diversification Index (ILDI): formative index calculated using the Herfindahl-Hirschman formula $(1 - \sum pi^2)$ [30] based on the relative shares of agricultural, extractive, and non-agricultural income. It will not be evaluated by Cronbach's alpha but described statistically.

- Climate Adaptation Practices Index (CAPI): reflective index, constructed from the standardized sum of reported adaptation practices (e.g., irrigation, resistant varieties, collective work). Internal consistency will be assessed by Cronbach's alpha [31] ($\alpha \geq 0.7$) and, when appropriate, by confirmatory factor analysis.

- Bioeconomy Engagement Index (BEI): formative ordinal variable, with increasing levels of engagement in bioeconomy value chains (0: no participation; 1: primary production; 2: processing; 3: processing + certification/market differentiation). Higher values indicate greater household-level involvement. For multilevel analyses, the BEI will be used primarily as a level-1 predictor (household-level) to examine associations with income, adaptive capacity, and adoption of mitigation/adaptation practices. To explore community-level patterns, the BEI will also be summarized at the settlement level using the mean or distribution of household scores (e.g., proportion of households at each engagement level). These aggregated measures will be used as level-2 contextual variables when relevant for hypotheses concerning settlement-level dynamics (e.g., HP5).

- Climate Vulnerability Perception Scale (CVPS): reflective scale, constructed as the average of Likert-type items (1–5) capturing perceptions of climate risk (drought, flood, crop loss, food insecurity). Internal consistency will be assessed by Cronbach's alpha ($\alpha \geq 0.7$) and unidimensionality by CFA.

- Municipal Climate Vulnerability Index (MCVI): formative index based on indicators of exposure (climate), sensitivity (e.g., agricultural dependence, poverty), and adaptive capacity (e.g., education, policy access). It will be constructed from standardized indicators (z-scores) and aggregated through PCA. PCA adequacy indicators (*KMO* > 0.6) and variance explained by the first component will be reported.

Each index will be statistically validated, described in terms of distribution, central tendency, and dispersion, and interpreted according to its reflective or formative nature.

**Qualitative primary data analysis.** Transcriptions of interviews and focus groups will be analyzed using Braun and Clarke's Reflexive Thematic Analysis [46] in six stages: (1) familiarization; (2) generation of initial codes; (3) search for themes; (4) review of themes; (5) definition and naming of themes; and (6) report production. This analysis will identify central themes related to local perceptions, barriers, facilitators, and understandings of climate change, adaptation, and bioeconomy.

All qualitative findings will be reported according to the COREQ guideline (or SRQR, where applicable), ensuring comprehensive documentation of data collection and analytic procedures.

**Integration and triangulation.** In the final phase, the findings will be integrated following a sequential explanatory mixed-methods design (QUAN→QUAL) and a joint display triangulation framework [41]. The integration will be reported in accordance with the Good Reporting of a Mixed Methods Study (GRAMMS) framework and the Standards for Reporting Qualitative Research (SRQR) guidelines for the qualitative component.

A triangulation matrix will be constructed to systematically map convergences, complementarities, and divergences between quantitative, qualitative, and secondary data findings for each research objective and hypothesis. The matrix will define decision rules for interpreting relationships: (I) Convergence: Findings from different methods provide corroborating evidence for the same conclusion; (II) Complementarity: Findings from different methods elucidate distinct, non-overlapping aspects of a phenomenon, together providing a richer, more complete picture; (III) Discordance: Findings from different methods are contradictory or inconsistent. In such cases, we will re-examine the data and the assumptions of each method to propose potential explanations for the discrepancy (e.g., differences in perspective between managers and farmers, social desirability bias in surveys). (IV) Silence: An expected finding from one method is absent in the data from another method.

For example, a quantitative pattern of low adoption of a bioeconomy practice (QUAN) will be examined considering reported barriers in interviews (QUAL) and the (non)existence of supportive policies identified in secondary data. This synthesis will enable not only the identification of *what* occurs but also *why* and *how* adaptation and resilience processes unfold.

**Handling of missing data.** The strategy for handling missing data will be determined by its mechanism and extent. During data collection, digital forms with mandatory field checks will minimize inadvertent missingness. Interviewers will be trained to probe for reluctant or "don't know" responses. Prior to analysis, we will generate a missingness pattern report. Little's Missing Completely at Random (MCAR) [47] test will be used to inform the choice of handling method, though we will primarily assume data is Missing at Random (MAR).

Our primary analysis will use a Full Information Maximum Likelihood (FIML) estimation within the mixed-effects models (lme4 package in R). FIML uses all available data points without discarding partial cases and provides unbiased estimates under the Missing at Random (MAR) assumption.

To assess the robustness, we will conduct sensitivity analyses using Multiple Imputation (mice package in R). We will create 20 imputed datasets, including all model variables and auxiliary variables correlated with missingness. The multilevel structure will be respected during imputation using the 2lonly method, incorporating cluster means of level 1 variables as predictors. All model variables, plus auxiliary variables correlated with missingness (e.g., interview duration, interviewer ID), will be included in the imputation model. Results from the pooled imputed models will be compared with the primary FIML analysis.

In addition to primary data, we will assess and report missingness in all secondary datasets, including climate indicators, socioeconomic data, municipal plans, and budget information. For missing values in external data sources, we will apply appropriate strategies depending on the mechanism: (i) triangulation with alternative official databases when equivalent indicators exist, (ii) imputation of continuous variables using multilevel or time-series–based methods when justified, and (iii) explicit coding of structural missingness (e.g., municipalities with no climate plans or no identifiable climate-related expenditures). The extent and implications of missing secondary data will be documented in the Municipal Climate Vulnerability Index (MCVI) and in the construction of the Municipal Climate Engagement Typology (MCET), and sensitivity checks will be conducted to assess the robustness of results to these gaps.

## Data management plan

All secondary data comes from public sources and will be fully documented. Primary surveys and qualitative data will undergo rigorous anonymization to protect participant confidentiality. Anonymized datasets, metadata, and analysis scripts will be deposited in the Open Science Framework (OSF) under a temporary embargo; raw qualitative data (audio or identifiable transcripts) will remain confidential as required by ethics guidelines. Detailed documentation will accompany all shared materials to ensure transparency and reproducibility.

## Study status and timeline

A pilot study was conducted in Marabá (PA) to refine instruments and procedures. Large-scale data collection is scheduled for December 2025 to December 2026, followed by data analysis and dissemination activities in 2027. These timelines reflect logistical constraints of fieldwork in remote Amazonian settlements.

## Discussion

The proposed protocol advances the literature by integrating climate resilience indicators with bioeconomy dynamics in agrarian reform settlements, an approach that has not yet been systematically applied in international research. By combining multilevel quantitative analysis with grounded qualitative evidence, the study addresses adaptation processes as they unfold across households, communities, and municipal institutional contexts. This integrated perspective reflects both the novelty and the global relevance of the proposed design.

The mixed-methods and multilevel analytical strategy was adopted in recognition of the fact that climate adaptation and bioeconomy practices are simultaneously shaped by household characteristics, collective organization, and local governance capacity. A purely quantitative approach would be insufficient to capture the narratives, meanings, and institutional mechanisms that condition adaptive behavior, while a purely qualitative design would limit systematic comparison across municipalities and settlements. The sequential explanatory design (QUAN→QUAL), supported by triangulation, therefore provides a coherent framework for addressing research questions. Alternative designs, including longitudinal panel studies [48] and experimental or quasi-experimental policy evaluations [49], were considered but proved infeasible due to the lack of comparable baseline data, constraints on randomization, and the logistical challenges of sustained fieldwork in remote Amazonian territories.

At the same time, the reliance on secondary data introduces constraints related to data availability, variable quality, and heterogeneity in reporting standards across municipalities. Climate indicators, budget classifications, and policy documents differ in completeness and temporal coverage. These limitations are addressed through triangulation across multiple official databases, the use of imputation and sensitivity analyses for incomplete indicators, and the explicit treatment of structural missingness, such as municipalities without formal climate plans or identifiable climate-related expenditures.

The qualitative component is also subject to potential social desirability and biases, particularly in interviews involving institutional actors or community leaders. This risk will be mitigated through careful interviewer training, rapport-building

strategies, and clear communication regarding confidentiality and the absence of normative expectations. The involvement of community-based researchers familiar with local contexts further strengthens trust and reduces outsider bias.

The intentional selection of municipalities and settlements necessarily limits the statistical generalizability of the findings to the entire state of Pará. However, the study does not seek to produce representative estimates; rather, it aims to generate analytically rich insights into key socioecological contexts. The diversity of selected municipalities and the transparent documentation of selection criteria allow readers to assess the transferability of findings to similar rural territories across the Eastern Amazon and beyond.

Beyond these constraints, the protocol contributes to existing international debates by bridging strands of literature that are often treated separately. While existing studies have examined connections between climate adaptation and rural livelihoods [50], as well as between rural livelihoods and bioeconomy pathways [51], the simultaneous integration of these three dimensions within a multilevel analytical framework, supported by grounded qualitative evidence, remains limited [52]. The adaptation of global frameworks – such as livelihood diversification, adaptation portfolios, and municipal climate governance – to the institutional realities of agrarian reform settlements addresses an important gap in the literature on rural socioecological transitions [53].

Finally, the applied orientation of the study reinforces the appropriateness of the chosen design. The findings are expected to inform municipal and state policymakers in Brazil by supporting the refinement of climate adaptation and bioeconomy strategies, public resource allocation, and territorial program design. With appropriate contextual adjustments, the methodological approach is also transferable to other regions characterized by decentralized climate governance, rural poverty, and socio-biodiversity-based economies.

Overall, while the study faces limitations inherent to its observational and context-specific nature, the mixed-methods, multilevel design – combined with transparency, triangulation, and sensitivity analyses – provides a robust framework for advancing empirical knowledge on climate resilience and bioeconomy pathways in vulnerable rural regions. The implications of these limitations will be explicitly assessed and clearly communicated to ensure that conclusions accurately reflect the scope and strength of the evidence.

## Supporting information

**S1 File. Household survey questionnaire.**
(PDF)

**S2 File. Focus group discussion guide.**
(PDF)

**S3 File. Semi-structured interview guide for public managers.**
(PDF)

## Acknowledgments

The authors would like to express their sincere gratitude to the rural communities and public managers of the state of Pará who, with their availability and insights, were fundamental to the data collection for this study.

## Author contributions

**Conceptualization:** Daniel Silva.

**Data curation:** Vitor Castro.

**Formal analysis:** Larissa Alves, Emilio Mendes.

**Funding acquisition:** Daniel Silva.

**Investigation:** Larissa Alves, Naurinete Reis, Maclem Erane Gonçalves dos Santos, Emilio Mendes.

**Methodology:** Daniel Silva, Larissa Alves, Naurinete Reis, Maclem Erane Gonçalves dos Santos, Emilio Mendes.

**Project administration:** Daniel Silva.

**Software:** Vitor Castro.

**Supervision:** Daniel Silva.

**Validation:** Naurinete Reis.

**Visualization:** Emilio Mendes.

**Writing – original draft:** Daniel Silva.

**Writing – review & editing:** Daniel Silva, Larissa Alves, Naurinete Reis, Maclem Erane Gonçalves dos Santos, Vitor Castro, Emilio Mendes.

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
