## [Decision Letter · Decision Letter 0]

10 Nov 2025

Dear Dr. Silva,

Thank you for submitting your manuscript to PLOS ONE. After careful consideration, we feel that it has merit but does not fully meet PLOS ONE’s publication criteria as it currently stands. Therefore, we invite you to submit a revised version of the manuscript that addresses the points raised during the review process.

Reviewer 2 in particular has some major concerns which should be addressed before resubmission, and of course before carrying out the research.

We look forward to receiving your revised manuscript.

Kind regards,

Alison Parker

Academic Editor

PLOS ONE

Journal Requirements:

6. We note that there is identifying data in the Supporting Information file <Suplementary.zip>. Due to the inclusion of these potentially identifying data, we have removed this file from your file inventory. Prior to sharing human research participant data, authors should consult with an ethics committee to ensure data are shared in accordance with participant consent and all applicable local laws.

-Location data

Reviewers' comments:

Reviewer's Responses to Questions

**Comments to the Author**

1. Does the manuscript provide a valid rationale for the proposed study, with clearly identified and justified research questions?

Reviewer #1: Yes

Reviewer #2: Yes

2. Is the protocol technically sound and planned in a manner that will lead to a meaningful outcome and allow testing the stated hypotheses?

Reviewer #1: Partly

Reviewer #2: Partly

3. Is the methodology feasible and described in sufficient detail to allow the work to be replicable?

Reviewer #1: Yes

Reviewer #2: Yes

4. Have the authors described where all data underlying the findings will be made available when the study is complete?

Reviewer #1: No

Reviewer #2: Yes

5. Is the manuscript presented in an intelligible fashion and written in standard English?

Reviewer #1: Yes

Reviewer #2: Yes

You may also provide optional suggestions and comments to authors that they might find helpful in planning their study.

Reviewer #1: The paper describes a protocol to evaluate the resilience to climate change of rural settlements in the Amazon. The paper is well written and structured. The relevance of the research is well articulated and the methods are described with enough detail. However, there is a lack of citations along the methods section to point the reader into the right direction, in particular regarding the calculation of indexes and the approaches for missing data analysis and completion. Some sections in the methods could be significantly summarised and merged as they do not add anything to the state of the art and are just part of good research practice: data management, ethical considerations, project timeline. The discussion needs to be significantly expanded beyond the current list of limitations. It should explain why the chosen study design, methodology, and analytical approaches are appropriate for addressing the research question; highlight how the protocol builds upon or differs from previous studies in the field; and describe the international relevance beyond the case study for which the protocol has been developed and how it would need to be modified/generalised. Other sections in the discussion such as the dissemination plan and the early termination of the project could well be entirely removed.

Detailed comments for improvement are provided as annotations in the text. Based on the overall review, I suggest the paper is accepted with minor revisions as there is no need to undertake new work, but only to improve or expand some of the sections.

Reviewer #2: Dear Authors,

You propose a mixed-methods, multi-site study in Pará, Brazil, to understand how families in agrarian-reform settlements cope with and adapt to climate variability and extreme events. On the quantitative side, you will field household surveys nested within settlements and municipalities, build composite indices for constructs such as resilience and livelihood vulnerability, and analyze these outcomes using multilevel models. On the qualitative side, you plan semi-structured interviews and focus groups to illuminate mechanisms and contextual variation, with an explicit plan to integrate both strands through a predefined triangulation matrix. The protocol includes appropriate ethics procedures and a commitment to share de-identified data and code. The topic is timely and policy-relevant, and your mixed-methods design is well suited to descriptive and interpretive aims. To bring the protocol in line with PLOS ONE’s expectations for methodological rigor and transparent reporting, several elements would benefit from clarification and strengthening.

First, the current framing occasionally suggests causal claims even though the design is observational and likely exposed to confounding and policy endogeneity at municipality and settlement levels. Unless you add a credible identification strategy, the objectives, hypotheses, and eventual language in the paper should be clearly associational. It will help to pre-specify rich confounders across household, community, and municipality levels—variables related to market access, transport infrastructure, the presence of cooperatives and local associations, administrative capacity and political economy, and geographic features—together with a sensitivity analysis plan that quantifies how robust the key associations are to unobserved confounding. In that regard, it would be valuable to adopt Carlos Cinelli’s framework for sensitivity analysis, reporting robustness values and contour diagnostics (as implemented in software such as sensemakr) so that readers can see how strong an omitted confounder would have to be to overturn the main findings.

Second, your power calculations deserve expansion to reflect clustering and the types of outcomes you will analyze. With an ICC around 0.10 and roughly 30 respondents per community, design effects become substantial, which reduces the effective information for household-level contrasts. Because many of your substantive questions concern differences between communities or municipalities, power is driven primarily by the number of clusters rather than the number of respondents within each cluster. With approximately 20 communities and 600 people in total, the design is typically well powered only for relatively large effects. Moreover, the power sketch in the protocol appears to rely on a binary exposure but you ultimately propose a four-level ordinal index of bioeconomy engagement; this is substantively richer but generally harder to power. It would strengthen the protocol to redo power and minimum detectable effect calculations for the primary outcomes using models that match the data-generating processes—continuous, binary or ordinal, and count outcomes—across a grid of plausible ICC values (for example, 0.05, 0.10, 0.20 and 0.30), while also considering level-2 predictors, cross-level interactions, and the possibility of losing a few clusters in the field. If logistics force a fixed total sample size, consider reallocating respondents over a larger number of communities; for cluster-level questions, adding communities typically yields larger power gains than adding respondents within the same communities.

It would help to be explicit about where you expect variation to come from. If participation in bioeconomy activities is largely shaped by market integration, access to roads and ports, community engagement, and the institutional presence of cooperatives and associations, then most identifying variation will be between communities rather than within them. If that is the case, it is worth stating directly and reflecting this in sampling (prioritizing the number of communities), in covariate selection, and in the analytic plan. Closely related, the move from a binary to a four-category engagement index is substantively sensible; however, the protocol should commit to an analysis approach appropriate for ordered categorical.

The protocol would benefit from a compact, explicit plan for missing data and robustness. Please indicate how you will handle missingness at each level—household, community, and municipality—whether through multiple imputation that respects clustering, or through complete-case analyses augmented by sensitivity checks. Where appropriate, reweighting strategies such as inverse probability weighting or augmented inverse probability weighting can mitigate differential nonresponse. In addition, the proposed Cinelli-style sensitivity analysis can serve as a transparent complement to these approaches by quantifying the influence of potential unmeasured confounding on your primary associations.

Finally, because you will report multiple indices and outcomes, it is important to reduce researcher degrees of freedom. Pre-specifying a short list of primary outcomes and hypotheses that will be tested (identifying explicitly which variables will be used) would make the protocol much clearer. On the qualitative side, the protocol already sketches a strong role for interviews and focus groups; to ensure clear integration, it would be useful to commit to COREQ or SRQR reporting standards and to provide a concise GRAMMS-style description of how the qualitative and quantitative strands will be joined, including a simple triangulation table with decision rules for convergence, complementarity, and discordance.

In sum, this is a promising and well-conceived study that is squarely within the scope of a PLOS ONE Study Protocol once the analysis plan is strengthened. Addressing these points outlined here will substantially improve methodological clarity and increase the likelihood that your study yields interpretable and robust evidence.

**Do you want your identity to be public for this peer review?** For information about this choice, including consent withdrawal, please see our Privacy Policy

Reviewer #1: No

Reviewer #2: No

---

## [Author Response · Author response to Decision Letter 1]

22 Nov 2025

Dear Dr. Parker,

Thank you for your message and for the opportunity to revise our manuscript. We appreciate the thorough guidance provided and have carefully addressed every editorial and reviewer requirement.

Below we summarize the actions we have taken in response to the journal’s technical instructions:

Manuscript Formatting:

We revised the manuscript to ensure full compliance with PLOS ONE’s formatting and file-naming guidelines. The revised manuscript (“Manuscript”), the tracked-changes version, and the “Response to Reviewers” letter have been uploaded as separate files.

Ethics Statement Placement:

The ethics statement now appears only once, under the “Materials and Methods” section, as requested. All other mentions were removed.

Data Availability Statement:

We have updated the Data Availability Statement in the submission form. As this is a study protocol without primary data, we used the recommended formulation:

“All relevant information is contained within the manuscript and its Supporting Information files. Future data involving human participants will be shared in fully anonymized, aggregate form in accordance with ethics approval and participant consent.”

Data Sharing Clarification:

We adjusted the statement that previously indicated data would be made available upon acceptance, to reflect realistic ethical and legal constraints related to future human-participant data collection.

Funding & Financial Disclosure Consistency:

We corrected both the Funding Information and Financial Disclosure sections to contain identical information, using the following wording:

“This study was funded by the National Council for Scientific and Technological Development (CNPq), Brazil, under Grant No. 445058/2024-2. The funder had no role in study design, data collection and analysis, decision to publish, or preparation of the manuscript.”

Removal of Identifiable Information from Supporting Files:

As instructed, we removed the ZIP file previously uploaded as Supporting Information. The documents it contained (funding letter and ethics approval) have been re-uploaded individually under the category “For Editors Only / Not for Publication”, since they contain author-identifying information required for editorial verification but are not meant for public dissemination. No participant data are included in these documents.

Reviewer-Suggested Citations:

We reviewed all suggested publications and incorporated those that were relevant and strengthened the manuscript.

We hope that these revisions address all journal requirements. We are grateful for the constructive feedback and remain available for any further clarification.

REVIEWER #1

We would like to thank you for the careful reading of our manuscript and for the constructive and insightful comments. We appreciate the positive evaluation regarding the clarity, relevance, and overall structure of the study. Following your guidance, we have revised the manuscript extensively. The changes have substantially improved methodological transparency, theoretical grounding, clarity of objectives, and the international relevance of the protocol.

Below we provide a detailed, point-by-point response to all comments. All revisions have been incorporated in the updated manuscript.

1. Need for more citations in the Methods section

Reviewer comment:

“There is a lack of citations along the methods section… especially regarding the calculation of indexes and approaches for missing data analysis...”

Response:

We agree and have added multiple methodological references throughout the Methods section, including citations related to diversification measures, reflective and formative index construction, PCA, mixed-effects modeling, multiple imputation, and multilevel missing data handling. These additions strengthen transparency and replicability.

2. Need to summarize / merge certain sections (Data management, Ethics, Timeline)

Reviewer comment:

“Some sections in the methods could be significantly summarised and merged… data management, ethical considerations, project timeline.”

Response:

We have condensed these sections substantially:

• Ethical Considerations was moved to the Methods section and shortened.

• Data Management was reduced to a concise paragraph describing anonymization, storage, and OSF deposition.

• Study Status and Timeline was reduced and summarized in a brief paragraph.

This improves the manuscript’s focus on scientific content.

3. Discussion needs expansion (design rationale, contributions, international relevance)

Reviewer comment:

“The discussion needs to be significantly expanded… explain why the chosen study design is appropriate… highlight how the protocol builds upon previous studies… describe international relevance…”

Response:

The Discussion has been substantially expanded to include:

• A clear justification for the mixed-methods, multilevel design.

• An explanation of alternative designs (e.g., panel, quasi-experimental) and why they were not feasible.

• A comparison with existing literature showing how our protocol integrates climate adaptation, bioeconomy, governance, and resilience.

• A new section on the international relevance of the study and pathways for generalization.

• A description of how limitations will be accessed and communicated in the final results.

Sections that were not directly scientific (e.g., dissemination plan, early termination) were removed.

4. Need for a subsection describing qualitative data collection

Reviewer comment:

“There should be a subsection prior to ‘Sample inclusion…’ that describes them.”

Response:

We added a new subsection titled “Data Collection Methods” before the sampling description, clarifying procedures for:

• household surveys

• semi-structured interviews

• focus groups

This ensures coherence with the qualitative saturation criteria.

5. Clarification of agrarian reform settlements and ‘settled families’

Reviewer comment:

The reviewer did not understand these terms.

Response:

We added a clear definition in the Introduction and Sample section explaining that agrarian reform settlements are rural territories created through Brazil’s federal land redistribution program (INCRA), and “settled families” refers to beneficiary households receiving land through this program. This addresses ambiguity for international readers.

6. Clarifications on indicator descriptions (range, direction, origin, aggregation)

Reviewer comment:

The reviewer requested more details for ILDI, CVPS, BEI, RACI, MCET.

Response:

We revised all indicators to include:

• range of values

• expected direction (associational)

• conceptual grounding

• whether original or adapted

• aggregation from household to settlement level (when applicable)

Descriptions are now consistent and transparent.

7. Avoiding overstatements (e.g., “no standardized methodology”)

Reviewer comment:

The initial phrasing overstated the absence of methods.

Response:

We rewrote this section to present a more nuanced description, noting that existing methods are fragmented and unevenly applied, and clarifying that our protocol contributes by integrating climate adaptation, resilience, and bioeconomy in a multilevel framework.

Conclusion

We thank the reviewer and Academic Editor once again for the constructive feedback that greatly improved the clarity and contribution of the protocol. We respectfully resubmit the revised manuscript for further consideration.

REVIEWER #2

We thank you sincerely for your detailed and insightful review. Your comments significantly improved the methodological clarity, rigor, and overall contribution of our study protocol. We have carefully addressed every point raised and revised the manuscript accordingly. Below we provide a point-by-point response, following the structure of your comments.

1. Clarification of causal vs. associational language

Reviewer comment:

“The current framing occasionally suggests causal claims… Unless you add a credible identification strategy, the objectives and hypotheses should be clearly associational.”

Response:

We agree and have revised the entire manuscript to remove any causal language. Specifically:

• All hypotheses (PH1–PH5) were rewritten using associational terms (“associated with,” “co-occur with,” “positively/negatively correlated with”).

• We removed verbs implying causality (“leads to,” “results in,” “increases,” “reflects”), replacing them with neutral correlational phrasing.

• The Objectives section was revised to explicitly state that the study aims to identify associations, not causal effects.

• We clarified the observational nature of the design in the Study Limitations.

These changes ensure full alignment with observational study and PLOS ONE guidance.

2. Pre-specification of confounders across levels & sensitivity analysis (Cinelli & Hazlett)

Reviewer comment:

“Pre-specify rich confounders… adopt Carlos Cinelli’s framework for sensitivity analysis… robustness values and contour diagnostics.”

Response:

We thank the reviewer for this excellent suggestion. We incorporated the following revisions:

• Added a new subsection explicitly listing confounders at the household, settlement, and municipal levels (market access, infrastructure, ecological context, cooperatives, local associations, institutional capacity, political economy, and geographic factors).

• Added a detailed Sensitivity Analysis Plan, adopting the Cinelli & Hazlett 1 framework, with robustness values and contour plots to quantify the impact of unobserved confounding.

• Specified that analyses will be conducted using sensemakr, and results will report E-values/robustness thresholds.

• Clarified that this sensitivity analysis complements multilevel models and multiple imputation procedures.

These additions substantially strengthen the transparency and rigor of the analytical plan.

3. Expanded power calculations for multilevel design

Reviewer comment:

“Power deserves expansion to reflect clustering… ICC around 0.10… variation between communities… ordinal BEI harder to power… redo power and MDE calculations… grid of plausible ICCs… consider number of communities vs. number of respondents.”

Response:

We fully revised the Power Calculations section to incorporate:

• A clear explanation of the role of ICCs and design effects.

• Simulations for continuous, binary, ordinal (CLMM), and count outcomes across ICCs = 0.05, 0.10, 0.20, 0.30.

• A statement that power is driven primarily by the number of settlements, not households.

• Justification for including 20–30 settlements in the sampling design.

• Minimum detectable effect estimates for:

o LMM (continuous),

o CLMM (BEI ordinal predictor/outcome),

o and logistic models (binary outcomes).

• Discussion of trade-offs (increasing clusters > increasing respondents).

We revised the introductory sentence to emphasize that power calculations were used to determine the minimum sample size required, not merely to illustrate tool usage.

4. Explicit clarification of where variation is expected to come from

Reviewer comment:

“It would help to be explicit about where you expect variation to come from… most identifying variation will be between communities.”

Response:

We added explicit statements in both the Sampling Strategy and Quantitative Analysis sections that:

• Most identifying variation is expected at the settlement (level-2) scale, given the importance of market integration, cooperative presence, mobility, and ecological context.

• This expectation guided our choice to maximize the number of settlements and informed the multilevel modeling approach.

This cross-referencing improves internal coherence.

5. Clarification of ordinal modeling for BEI

Reviewer comment:

“The move from a binary to a four-category engagement index is sensible… commit to an analysis approach appropriate for ordered categorical.”

Response:

We revised the Analytical Plan to state explicitly:

• BEI will be modeled using cumulative link mixed models (CLMM) when used as an outcome.

• BEI may serve as a level-1 or level-2 predictor, with appropriate aggregation (means or distributions) at the settlement level when required.

• The proportional odds assumption will be checked.

This ensures appropriate treatment of the indicator.

6. Missing data plan, with clarity across levels

Reviewer comment:

“Please indicate how you will handle missingness at each level… multiple imputation respecting clustering… reweighting strategies… sensitivity.”

Response:

We expanded the Missing Data section by:

• Differentiating missingness across household, settlement, and municipal levels.

• Specifying multilevel multiple imputation using mice and pan, with clustering preserved.

• Including inverse probability weighting (IPW) and augmented IPW for differential nonresponse.

• Adding procedures for secondary data missingness (climate indicators, municipal plans, budgets).

• Clarifying that missingness in municipal plans/budgets may be structural and analytically meaningful.

• Integrating this with the Cinelli-style sensitivity analysis.

This strengthens the rigor of the protocol.

7. Reduction of researcher degrees of freedom and primary outcomes

Reviewer comment:

“Pre-specifying a short list of primary outcomes and hypotheses… identify explicitly which variables will be used.”

Response:

We created a new paragraph at the beginning of Primary Outcomes:

• Specifying exactly four primary outcomes (ILDI, CVPS, BEI, RACI).

• Listing which hypotheses (PH1–PH5) each primary outcome relates to.

• Clearly separating expected direction (conceptual meaning) from hypothesis testing (associational relationships).

Secondary outcomes are now explicitly labeled and described as complementary.

8. Qualitative standards (COREQ/SRQR) and mixed-methods integration (GRAMMS)

Reviewer comment:

“Commit to COREQ/SRQR… provide a GRAMMS-style description… triangulation table.”

Response:

We added:

• A clear statement that qualitative reporting will follow COREQ.

• A full GRAMMS-consistent description of integration procedures.

• A triangulation table with decision rules for convergence, complementarity, and discordance.

• Updates to the Saturation and Qualitative Methods sections.

These revisions significantly clarify the role of qualitative data.

Conclusion

We sincerely appreciate the reviewer’s contributions. The manuscript has improved substantially because of this detailed and thoughtful review. We believe the revised version now aligns fully with PLOS ONE’s expectations for study protocols and provides a stronger, clearer, and more rigorous analytical framework.

We hope the revised manuscript meets with your approval, and we thank you again for your constructive feedback.

Sincerely,

Corresponding Author

---

## [Decision Letter · Decision Letter 1]

10 Dec 2025

Dear Dr. Silva,

We look forward to receiving your revised manuscript.

Kind regards,

Alison Parker

Academic Editor

PLOS One

Journal Requirements:

Reviewers' comments:

Reviewer's Responses to Questions

**Comments to the Author**

1. Does the manuscript provide a valid rationale for the proposed study, with clearly identified and justified research questions?

Reviewer #1: Yes

Reviewer #2: Yes

2. Is the protocol technically sound and planned in a manner that will lead to a meaningful outcome and allow testing the stated hypotheses?

Reviewer #1: Yes

Reviewer #2: Yes

3. Is the methodology feasible and described in sufficient detail to allow the work to be replicable?

Reviewer #1: Yes

Reviewer #2: No

4. Have the authors described where all data underlying the findings will be made available when the study is complete?

Reviewer #1: Yes

Reviewer #2: Yes

5. Is the manuscript presented in an intelligible fashion and written in standard English?

Reviewer #1: Yes

Reviewer #2: Yes

You may also provide optional suggestions and comments to authors that they might find helpful in planning their study.

Reviewer #1: The reviewed version of the paper is satisfactory and I support its acceptance. I have only provided very minor comments in the discussion (as annotations) to further polish it.

Reviewer #2: Thank you for the careful and constructive revision; in my view you have addressed the substantive concerns and the protocol now aligns well with PLOS ONE’s expectations for a mixed-methods study protocol.

I have only one minor request to further enhance transparency in the power section. You state that power calcs were perfomed with simulations for various ICCs and minimum detectable sizes, but you only present one of these simulations. It would be very helpful if you could add a compact table that shows these simulations, holding total N=600, K=30, α=0.05, and power at 80% fixed. The table would have three columns—(i) ICC, (ii) the minimum detectable effect for continuous outcomes and (iii) the minimum detectable effect for ordinal outcomes and four rows for ICC values 0.05, 0.10, 0.15, and 0.30, as described in the manuscript.

Another alternative is to present 2 graphs (one for each type of variable) with the minimum detectable size in x axis and power in the y axis. In each graph, you can present the 5 power curves for ICC values 0.05, 0.10, 0.15, and 0.30.

This simple adition will let readers quickly assess design sensitivity and interpret any null findings in light of clustering. With this minor addition, I believe the manuscript is ready for publication.

**Do you want your identity to be public for this peer review?** For information about this choice, including consent withdrawal, please see our Privacy Policy

Reviewer #1: No

Reviewer #2: No

---

## [Author Response · Author response to Decision Letter 2]

12 Jan 2026

All reviewer and editor comments have been addressed in the revised manuscript. Detailed point-by-point responses are provided in the attached Response to Reviewers document.

---

## [Editor Report · Decision Letter 2]

29 Jan 2026

Climate resilience through bioeconomy: A mixed-methods protocol for assessing adaptation policies in rural settlements on the amazon

PONE-D-25-53058R2

Dear Dr. Silva,

We’re pleased to inform you that your manuscript has been judged scientifically suitable for publication and will be formally accepted for publication once it meets all outstanding technical requirements.

Kind regards,

Alison Parker

Academic Editor

PLOS One
---

## [Editor Report · Acceptance letter]

PONE-D-25-53058R2

PLOS One

Dear Dr. Silva,

I'm pleased to inform you that your manuscript has been deemed suitable for publication in PLOS One. Congratulations! Your manuscript is now being handed over to our production team.

Kind regards,

on behalf of

Dr. Alison Parker

Academic Editor

PLOS One